# Consumers’ Motives for Eating and Choosing Sweet Baked Products: A Cross-Cultural Segmentation Study

**DOI:** 10.3390/foods9121811

**Published:** 2020-12-07

**Authors:** Annchen Mielmann, Thomas A. Brunner

**Affiliations:** 1School of Physiology, Nutrition and Consumer Sciences, North-West University, Potchefstroom 2531, South Africa; 2Food Science and Management, School of Agricultural, Forest and Food Sciences (HAFL), Bern University of Applied Sciences, Länggasse 85, 3052 Zollikofen, Switzerland; Thomas.Brunner@bfh.ch

**Keywords:** food choice, motives, cross-cultural, cluster analyses, FCQ, MFES

## Abstract

This study aimed to examine consumers’ motives for eating and choosing sweet baked products (SBPs). A cross-cultural segmentation study on a South African (SA) and Swiss population sample (*n* = 216), was implemented using the Motivation for Eating Scale (MFES) and the Food Choice Questionnaire (FCQ). Cluster analyses provided three consumer segments for each population sample: the balanced and the frequenters for both countries, the deniers for SA, and the health conscious for Switzerland. South Africans liked SBPs more than the Swiss respondents, however the Swiss sample consumed SBPs more often. Environmental and physical eating were the most relevant motives when eating SBPs for the SA and Swiss group, respectively. For both samples, sensory appeal was the deciding factor when choosing SBPs. Cross-cultural studies of food choices are important tools that could help improve the current understanding of factors that influence the eating behavior of ultra-processed foods to promote healthy food choices through local and global perspectives. This paper highlights that more research is needed on consumers’ motives for choosing and eating ultra-processed foods in order to develop specific integrative cultural exchange actions or intervention strategies to solve the obesity issue.

## 1. Introduction

Consumers are rethinking the way that they select and eat food. The frequency of snacking and consuming snack foods is constantly growing across the globe [1,2]; the eating frequency of processed [3] and ultra-processed food has increased both at the household level and outside our homes. These shifts in the global food market combined with changes in consumers’ food choices have allowed demanding adjustments in the world food supply, resulting in dietary challenges. One example of this is the change to ultra-processed foods containing added sugars, refined grains, and carbohydrates [4]. Ultra-processed products are formulated primarily from artificial ingredients and typically contain additives and preservatives [5]; they cause nutritionally unbalanced diets and have been associated with obesity. Sweet baked products (SBPs), such as cake mixes and many varieties of cakes, pastries, biscuits, sweetened breads, and other sweet snack varieties, are classified as ultra-processed foods—these foods are typically high in sodium, saturated fat, and added sugar.

The food provisions of many high-income countries are dominated by ultra-processed food, for example Switzerland [6,7], and the manufacturing and eating of these products is swiftly expanding in middle-income countries [5,6] such as South Africa (SA) [8,9].

The term food choice motives refers to peoples’ motives for selecting or eating food [10]. Gaining insight into food choice motives is beneficial for promoting campaigns, innovations, strategies, and interventions related to food intake [11], as unhealthy food choices are among the leading risk factors for the development of obesity [12]. Researchers [13] have suggested the term cross-cultural interpretation for the interpretation of food choice motives—this means that there are discrepancies in norms and values between people, and cultural perception is best demonstrated through the study of consumers’ food choices. Within food choice motives, culture-specific differences can be applied by means of interventions to modify consumers’ food behavior across different cultures and populations, consuming healthier foods and cultivating wellbeing [14].

Researchers [15] have assessed the application of the (FCQ) [16] across cultures and identified only two food choice studies performed in Africa, of which one was bound to a student-based sample in SA [17] and the other was the validation of the FCQ in Cape Verde [18]. Therefore, based on the previous recommendations of cross-cultural research on food choice motives [19,20], this is a cross-cultural segmentation study on food motives comparing a European first-world country and an emerging African country. Considering the two countries’ distinctive food choice culture, we predict that we will find differences. This cross-cultural study will help to not only improve the understanding of consumers’ motives for their choices of SBPs but will also help to identify specific segments of the population that should be prioritized in order to change consumers’ motives for choosing and eating healthier foods.

The aim of this paper was to examine consumers’ motives for eating and choosing SBPs by conducting a cross-cultural segmentation study on a South African and Swiss population sample, implementing the Motivation for Eating Scale (MFES) [21] and the Food Choice Questionnaire (FCQ) [16].

## 2. Materials and Methods

### 2.1. Sampling and Data Collection

A non-probability purposive sampling method allowed access to the constricted population in this study. A total of 216 adult consumers (77 men and 139 women) from South Africa and Switzerland participated in the study. The SA sample consisted of 106 respondents (26 males and 80 females) from the Gauteng Province, and the Swiss sample comprised of 110 respondents (51 males and 59 females) from the German-speaking region of Switzerland.

For data collection, the respondents first received a link to an electronic screening questionnaire. Consumers who were mainly responsible for purchasing and preparing food were requested to complete the questionnaire. The screening questionnaire confirmed whether the respondents were: (1) a citizen of the specific country, (2) older than 18 years, and (3) consumers of SBPs. The respondents who qualified for the study then received a link to the main electronic questionnaire. The survey was administered using QuestionPro (v20.4, Seattle, DC, USA). In both countries, data collection occurred from December 2017 to February 2018.

All the respondents were required to give informed consent and participation in this study was completely voluntary. The respondents were reassured of their confidentiality and the privacy of the information that they supplied. The Scientific Committee of School of Agricultural, Forest, and Food Sciences, Bern University of Applied Sciences, approved the study (2016.08).

### 2.2. Questionnaire

All the questions were translated from English to German and checked by a native German speaker. A qualified language translator back-translated the questionnaire to English. The translators and authors of the paper discussed the differences between the original wording and the back-translation of the items until there was agreement that the German and English versions had the same meaning. The questionnaire consisted of four sections. A summary of the scale and items of sections one to three of the questionnaire is presented in Table 1.

The first section included three questions to determine consumers’ (1) liking of SBPs by means of a 4-point Likert-type scale (1 = not at all; 2 = to a small extent; 3 = to some extent; 4 = very much); (2) frequency of SBP consumption using a 5-point Likert-type scale (1 = never; 2 = once a year/rarely; 3 = monthly; 4 = weekly; 5 = daily); (3) awareness of sugar intake by means of a 5-point Likert- type scale (1 = not at all; 2 = to a small extent; 3 = to some extent; 4 = to a great extent; 5 = I don’t know) [22].

In the second section, the Motivation for Eating Scale (MFES) was employed to measure consumers’ main motives for consuming SBPs by means of a 5-point Likert-type scale (1 = never; 2 = almost never; 3 = sometimes; 4 = often; 5 = always) [21,23,24]. The respondents were asked to complete the phrase: “The situations or conditions that most often exist when I eat sweet baked products are when I...”

This questionnaire was comprised of four eating subscales: environmental, social, physical, and emotional [21]. For this study, only questions from subscales in the MFES relating to food choice were used; these were the initiation of the eating and the how people decide what to eat subscales.

In the third section, the respondents’ food choices were determined with the Food Choice Questionnaire (FCQ) [16] using 5-point Likert-type statements (1 = never; 2 = almost never; 3 = sometimes; 4 = often; 5 = always). The respondents were asked to complete the phrase: “I choose sweet baked products because it…” This FCQ aimed to determine the respondents’ food choices motives: mood, health, sensory appeal, convenience, price, weight control, and natural content [16]. The FCQ are context-dependent but the findings indicate variation in the food choice motives when the study acquires samples from various cultures [25]. Therefore, it will be relevant to determine consumers’ food choice motives for the two cultural groups because of this variation.

The fourth section of the questionnaire, presented in Table 2, included demographic questions, including gender, age, level of education, home language, marital status, monthly household income, occupation, and place of residence. Respondents had the option not to disclose their monthly income.

### 2.3. Data Analysis

Cronbach’s alpha coefficient was used to test the internal consistency of the constructs: (1) liking for SBPs; (2) frequency of SBP consumption; (3) awareness of sugar intake; (4) motives for eating (physical, emotional, and environmental); (5) food choice motives. Satisfactory findings were presented for all the scales (Cronbach’s alpha 0.72–0.91) (Table 1).

The means (of all the scales) were calculated and used for a hierarchical cluster analysis. Ward’s method was applied using the squared Euclidean distance for each country separately. The use of an agglomeration schedule and calculating the percentage of change in the clustering coefficients were applied and evaluated for results between two and eight clusters. The largest percentage increases occurred when the five into four clusters and three into two clusters were combined; therefore, the five- and three-cluster results were subjected to general linear model (GLM) analyses followed by contrast analyses. Robust tests (Brown–Forsythe and Welch) were applied due to the unequal cluster sizes and heteroscedasticity. Using contrast analyses, the results indicated that each cluster emerged significantly for the three-cluster solution for the SA and Swiss sample (Table 3). Therefore, the three-cluster solution was chosen for further analyses. GLM analyses, such as robust tests (Brown–Forsythe and Welch), were applied to determine the significance of the demographic variables (Table 2), the respondents’ liking and frequency of consumption of SBPs (Table 3); consumers’ motives for eating and choosing SBPs (Table 4); and the BMI score, frequency of exercise, and sleep per week (see Appendix A
Table A1 for results) between the three clusters. Statistical analyses were performed with IBM SPSS Statistics 24.

## 3. Results and Discussion

### 3.1. Description of the Sample

The South African respondents’ average age was 40.3 years; 24.1% were under 35 years of age and 81.9% were above the age of 35 years (Table 2). The Swiss respondents’ average age was 54.5 years; 25% were under 35 years of age and 85% were above the age of 35 years. The majority of the sample consisted of women living with a partner. The majority of the SA sample (94.3%) was employed, while most of the Swiss respondents (69.1%) were not employed or were on pension. For the disclosure preference for the SA (12.5%) and Swiss respondents’ (23.6%) monthly household income, a difference of more than 10% was evident. A description of the demographic results of the samples is presented in Table 2.

### 3.2. Description of the Clusters

The cluster analyses provided three segments with definite attitudes towards SBPs for SA (*n* = 106) and Switzerland (*n* = 110), respectively. The first segment, comprising 43.4% and 31.8% of the sample, respectively, of the SA and Swiss respondents, was described as the balanced; the second segment, 34.9% and 38.2% of the sample, respectively, for the SA and Swiss groups, was called the frequenters. A third segment was present for both groups; this segment was termed the deniers for the SA group (21.7%) and the health conscious for the Switzerland (30%) group. Respondents’ liking for and frequency of consumption of SBPs and their awareness of sugar intake are provided in Table 3. Table 4 indicates the respondents’ motives for choosing to eat SBPs.

The findings show that there are significant differences within the SA and Switzerland clusters. For the South African population, significant differences are present for the affinity for SBPs, weight control (with respect to motives for choosing) and their awareness of their sugar intake. For the Swiss group, significant differences are noticeable for environmental eating (with respect to their motives for eating) and convenience (with respect to their motives for choosing). For both population groups, significant differences are visible for frequency of SBP consumption, emotional eating (with respect to motives for eating) and their motives for choosing: natural content, price, mood, sensory appeal and health. When comparing the two population samples, South Africans liked SBPs more than the Swiss participants, however the Swiss respondents consumed SBPs more often. The results from the study population for both countries suggest that consumers were at least somewhat aware of their intake of sugar in their diets. A brief description of each segment is provided.

#### 3.2.1. SA: The Balanced (43.4%)

The balanced group had the highest average age and the largest share of single and employed respondents of all the SA Segments. Although their awareness of sugar intake was low, they tended to eat SBPs less frequently. For these respondents, emotional eating and sensory appeal were the most relevant motives when eating and choosing SBPs.

#### 3.2.2. Swiss: The Balanced (31.8%)

This segment had the highest average age and the largest share of females of all the Swiss segments. These consumers were highly educated and most of them no longer worked. They indicated the greatest liking for SBPs but consumed them the least frequently. The balanced consumers regarded physical eating and sensory appeal as important motives for eating and choosing SBPs.

#### 3.2.3. SA: The Frequenters (34.9%)

Of all SA segments, the frequenters were the ones who ate SBPs the most often. This segment was mainly comprised of women; however, it also contained the largest group of male consumers earning a very high income and living with a partner. This groups’ motives for eating and choosing SBPs were influenced by physical eating and the food products’ sensory characteristics.

#### 3.2.4. Swiss: The Frequenters (38.2%)

This segment had the highest percentage of males of all the Swiss segments. The frequenters from Switzerland ate SBPs the most frequently. Of all Swiss segments, it was the segment with the highest percentage of employed people, the highest income, and the highest percentage of singles. The frequenters regarded physical eating and sensory appeal as important motives for choosing and eating SBPs.

#### 3.2.5. SA: The Deniers (21.7%)

This segment had the lowest average age and the highest percentage of females of all the SA segments. These respondents were highly educated. This group indicated the greatest affinity for SBPs as well as the greatest awareness of the intake of sugar in their diet. The deniers’ motives for eating were mainly influenced by environmental conditions, and sensory characteristics motivated them to choose SBPs.

#### 3.2.6. Swiss: The Health-Conscious (30%)

The health-conscious consumers were well educated. Although this group indicated a high frequency of SBP consumption, they were very aware of their dietary sugar intake. For these respondents, physical eating and sensory appeal were distinct motives for eating and choosing SBPs, respectively.

### 3.3. Motives for Eating SPBs: Environmental and Physical Eating

Environmental and physical eating were the most relevant motives when eating SBPs for the SA and Swiss group, respectively. Individual motivations for consuming foods have been categorized into three groups [26]: environmental (eating activated by an object in direct surroundings, such as odours), emotional (eating due to boredom, or other emotional situations), or physical (eating due to internal cues, such as a rumbling stomach). Consumers’ decisions or motives for eating SBPs may be influenced by the location of the eating environment, whether at home or out of the home (such as supermarkets, convenience stores, vending machines, takeaways, cafes, and restaurants), and environmental stimuli such as passing a fast food outlet or restaurant, advertising, preparing food, watching a movie, or standing in the checkout stand of a food retailer [27]. South African consumers living in rural and urban areas are growing less of their food and buying more—there is need for evidence to support the premise that consumers’ environments can be controlled to influence healthier choices. Altering the food environment (so that consumers’ food choices about what to consume default to healthier preferences) will be more beneficial to a population’s health than education. In addition, research is needed to assess the local food environment amongst populations [28]. National data from the food environment are needed to maintain campaigns. Efforts to improve healthier food choices and decrease diabetes, obesity, and other non-communicable diseases will reach more success when backed by initiatives that want to create healthier food environments [29].

Physical eating usually occurs in reaction to a physical hunger cue, such as a rumbling stomach or any noticeable sign that the body requires food [30,31]. Even in the absence of a biological cue, people may eat because the food is available. For example, in the case of unnecessary snacking, a non-habitual snacker lacks a biological motivation for eating snack foods, therefore snacking without hunger will elevate the amount of energy consumed, which can result in consequent weight gain [32,33]. Using a qualitative multivariate analysis (QMA) map, researchers investigated the snacking behavior of adults towards food from Australia and China and found that sweet biscuits were ranked as likely to eat everyday [34]. A study determined how often Swiss respondents usually ate breakfast [35]. Skipping breakfast could be correlated with overconsumption later in the day, due to the feeling of intense hunger followed by the ingestion of sugar-dense and high-fat snack foods. It was found that females with high-frequency snack consumption were more prone to skip breakfast and were less consciousness about their health when compared with females in the lowest snack frequency segment [36]. In addition, hunger also motivated food intake and seeking behavior [36]—it motivates purchases in a virtual environment and it increases the probability of selecting unhealthy or energy-dense food products. Therefore, hunger directly maximizes wanting for food or the incentive value, which in turn could evoke the consumption of unhealthy foods [37] such as SBPs. Researchers [38] examined the correlation between breakfast composition and abdominal obesity among regular breakfast eaters of Swiss respondents and found that consumers who eat breakfast regularly had less abdominal obesity if their daily breakfast was composed of yogurt, nuts/seeds, cereal flakes, and fruit. A meta-analysis confirmed that skipping breakfast is associated with overweight/obesity and skipping breakfast increases the risk of overweight/obesity [39]. Therefore, the implementation of interventions to prevent the skipping of meals is needed to encourage consumers to change their eating patterns and to promote the consumption of healthier foods to improve their health status.

### 3.4. Motives for Choosing SBPs: Sensory

For both samples, sensory appeal was the deciding factor when choosing SBPs, suggesting that the sensory features such as taste and smell were important to the respondents. In accordance with other studies, the results showed that European consumers rated sensory appeal highly [40,41]. A recent study found that taste was important for South Africans when choosing SBPs; for example, sweetness was a key marker of whether a treat was found to be satisfying and rewarding [42]. Energy-dense foods are highly likeable, because their sensory features will trigger the brain’s reward areas [43] and their intake is often paired with positive effects by means of direct consumption while experiencing positive emotions and indirectly through the marketing of these foods [44]. A greater affinity for energy-dense foods is correlated with consuming these foods more often [45]. Therefore, more research is needed to identify and develop new techniques and strategies when measuring eating behavior associated with ultra-processed food consumption. These strategies should then be applied to the design of food products that will increase consumers’ wellbeing. For example, a consumers’ sensory experience will motivate their choice—they will experience low-sugar products with a different taste than regular sweet-tasting products [46]. Taste entails perceptions, beliefs, and identity—therefore, there is need to investigate it as a relational and cultural entity [47].

### 3.5. Implications

The cross-cultural similarities and differences as presented in this paper suggest the development of a specific integrative cultural exchange action or intervention plan to solve the obesity issue. Evidently, the deniers and frequenters segment from SA and the frequenter segment from Switzerland need to be targeted, as we speculate that these consumer segments will escalate, which will increase the level and prevalence of obesity in these countries; however, the results of this paper should rather be integrated in a holistic approach to promote better health and wellbeing for all consumers by managing overweight and obesity within the target populations. Consumers are poor at estimating objective risks; they overestimate their capacity for self-control and underestimate the health risks associated with the choices they make [48]. Thus, there is a need to conduct comprehensive research on the associations between consumers’ environmental and physical motives for eating ultra-processed foods such as SBPs and their sensory appeal motives for choosing these foods to effectively address the high prevalence of overconsumption. Researchers and public health practitioners should delve intensively into methods such as nudging and conditioning as intervention strategies to change consumers’ motives for choosing and eating ultra-processed foods. These interventions should support the availability and accessibility of healthy food and health promotional activities.

### 3.6. Recommendations and Limitations

The findings from this paper are useful and provide meaningful information and suggestions for consumer scientists, food communities, health administrators, and the food industry to promote healthy food choices and educate consumers about healthy food alternatives [49]. However, more data on ultra-processed food choices should be collected from South Africa and Switzerland in order to develop a sustainable platform to inform consumers in their local food environments how to make healthier food choice decisions that will support their wellbeing. Implementing recommendations to improve the wellbeing of consumers may result in less overweight and obese consumers and better health and therefore a healthier society. Furthermore, more research could determine if the relative differences in demographic characteristics across the two samples will suggest different levels of affluence and exposure to ultra-processed foods.

The current study had some limitations that need to be addressed. The use of electronic surveys can interfere with obtaining a representative sample, because some socio-demographic groups may be unnecessarily excluded—this, in turn, can alter the generalizability of the results. Electronic surveys do not enable the researcher to control measures that are more manageable in usual face-to-face administration. Respondents from different countries had different mean ages, and it could be the case that SBP consumption as well as the respondents’ motives for choosing and eating SBPs vary as a function of age. The difference in samples sizes across the two countries and the overall small samples are additional limitations for this study. Increasing the sample sizes should be an objective for future research to increase the credence in the study findings.

## 4. Conclusions

Ultra-processed food choices are complex and continually changing. Due to each sample’s unique food culture and the significant differences in people’s motives for eating and choosing SBPs, our results indicate that environmental and physical eating were the most relevant motives when eating SBPs for the SA and Swiss group, respectively, and for both groups sensory appeal was the deciding factor when choosing SBPs. As there are several motivational factors that influence consumers’ food decisions, it is necessary to understand these factors contribute to an individual’s food choice and how they vary according to cultural aspects. Cross-cultural studies of food choices are important tools that could help improve the current understanding of factors that influence eating behavior, provide sustainable nutrition education, and ultimately promote healthy food choices through local and global perspectives.

## Figures and Tables

**Table 1 foods-09-01811-t001:** Items used for cluster analysis, including internal consistency analysis.

Items	Cronbach’s Alpha	Reference
Section 1		
Liking for sweet baked products	0.815	new
Cookies/Biscuits; Large cake/Cupcakes; Muffins; Snack bars; Pancakes/Waffles/ Flapjacks; Doughnuts; Brownies; Traditional		
Frequency of sweet baked products consumption	0.726	new
Cookies/Biscuits; Large cake/Cupcakes; Muffins; Snack bars; Pancakes/Waffles/ Flapjacks; Doughnuts; Brownies; Traditional		
Awareness of sugar intake	0.816	Boggiano (2016)
Do you think the consumption of sugar is unhealthy? Do you think the intake of sugar causes obesity? Do you think the intake of sugar causes diabetes? I am more concerned about the ingredients in sweet baked products than I was 3 years ago ^Δ^ I pay more attention to the amount of sugar added in a sweet baked product than I did 3 years ago ^Δ^ I am concerned about the amount of sugar in sweet baked products ^Δ^ I care about my sugar intake		
Section 2		
Motives for eating SBPS		Hawks et al. (2003)
Physical eating	0.734	
Need physical energy Feel physical hunger pains Am physically hungry and food sounds good Am weak/lightheaded because I haven’t eaten Have forgotten to eat and am starved		
Emotional eating	0.850	
Get bored Want to cheer up Feel irritable because I haven’t eaten Want to treat myself Feel it is connected to a memory of happiness ^†^ Once started to eat, it’s hard to stop ^†^ Overconsume when under stress Reward myself after a challenging task—I feel I “deserve” it ^†^		
Environmental eating	0.728	
Realise it’s mealtime, so I automatically eat Have tempting food in front of me Am busy preparing food See something good at a checkout stand See an advertisement of the product ^Δ^		
Section 3		
Motives for choosing SBPS		Steptoe et al. (1995)
Health	0.762	
Is nutritious Contains a lot of vitamins and minerals/ Is high in sugar ^Δ^ Keeps me healthy		
Mood	0.909	
Cheers me up Helps me cope with stress Keeps me awake/alert Helps me relax Makes me feel good Help me cope with life		
Convenience	0.856	
Is easily available in shops and supermarkets Takes no time to prepare/ Can be bought in shops close to where I live or work		
Sensory appeal	0.719	
Tastes good Smells nice Has a pleasant texture Looks nice		
Natural content	0.817	
Contains no additives Contains natural ingredients Contains no artificial ingredients		
Price	0.858	
Is not expensive Is good value for money Is cheap		
Weight control	0.805	
Is low in calories Is low in fat Helps me control my weight		

^Δ^ Items rephrased, ^†^ Items inspired from observations made by the cited source.

**Table 2 foods-09-01811-t002:** Demographic features by clusters of the SA and Swiss sample.

	SA:				Swiss:			
	Balanced	Deniers	Frequenters	Overall Sample	Balanced	Health-Conscious	Frequenters	Overall Sample
	43.4%	21.7%	34.9%	100.0%	31.8%	30.0%	38.2%	100%
	*n* = 46	*n* = 23	*n* = 37	*n* = 106	*n* = 35	*n* = 33	*n* = 42	*n* = 110
Gender ***								
Male	28.2%	8.7%	34.5%	24.5%	43.2%	45.8%	50.3%	46.4%
Female	71.8%	89.2%	65.5%	75.5%	56.8%	54.2%	49.7%	53.6%
Average age ***	40.5	40.0	40.1	40.3	58.3	52.8	52.8	54.5
Education level SA *								
Grade 12	23.9%	21.7%	21.6%	22.4%				
Certificate/Diploma	52.2%	32.6%	48.2%	44.3%				
Degree	9.6%	20.4%	6.8%	12.3%				
Post-graduate degree	14.1%	25%	23.2%	20.8%				
Education level Swiss *								
Baccalaureate school or below					31.4%	48.5%	45.2%	41.8%
Professional education					20.0%	24.2%	21.4%	21.8%
University of applied sciences					22.9%	24.2%	16.7%	20.9%
University/federal institute of technology					25.7%	3.0%	16.7%	15.5%
Monthly income SA *								
Less than R4000	4.3%	4.3%	2.7%	3.8%				
R4000–R8000	2.2%	8.7%	5.4%	5.4%				
R8001–R20,000	32.6%	26.1%	27.0%	28.6%				
R20,001–R50,000	26.1%	43.5%	29.7%	33.1%				
More than R50,0001	19.6%	8.7%	21.6%	14.45				
Disclosure preference	15.2%	8.7%	13.5%	12.5%				
Monthly income Swiss *								
CHF 6001–7500					2.9%	12.1%	16.7%	10.9%
CHF 7501–9000					11.4%	24.2%	7.1%	13.6%
CHF 9001–10,500					11.4%	6.1%	11.9%	10%
More than CHF 10,500					22.9%	15.2%	23.8%	20.9%
Disclosure preference					22.9%	30.3%	19.0%	23.6%
Marital status **								
Single/widow/ widower/divorced	47.3%	32.0%	36.7%	38.7%	14.6%	18.5%	24.1%	19.1%
Married/living with A partner	52.7%	68.0%	63.3%	61.3%	85.4%	81.5%	75.9%	80.9%
Occupational status ***								
Working	98.8%	92.3%	91.8%	94.3%	19.4%	35.8%	37.5%	30.9%
Not working/on pension	1.2%	7.7%	8.2%	5.7%	80.6%	64.2%	62.5%	69.1%

Note. * *p* < 0.05; ** *p* < 0.01; *** *p* < 0.001.

**Table 3 foods-09-01811-t003:** Mean scores on the clustering scales and contrast analysis results by clusters of consumers’ liking and frequency of consumption of SBPs.

	SA:			Swiss:		
	Balanced	Deniers	Frequenters	Balanced	Health-Conscious	Frequenters
	43.4%	21.7%	34.9%	31.8%	30.0%	38.2%
Liking of sweet baked products ***^,1^	3.51	3.75	3.45	3.24	3.03	2.95
Frequency of sweet baked products consumption **^,1;^ *^,2^	3.04	3.05	3.16	3.17	3.40	3.54
Awareness of sugar intake **^,1^	2.45	2.54	2.51	2.51	2.54	2.42

Note. * *p* < 0.05; ** *p* < 0.01; *** *p* < 0.001. ^1^ SA. ^2^ Swiss. Mean scores on Likert scales: Liking—1 = not at all; 2 = to a small extent; 3 = to some extent; 4 = very much. Frequency—1 = never; 2 = once a year/rarely; 3 = monthly; 4 = weekly; 5 = daily. Awareness—1 = not at all; 2 = to a small extent; 3 = to some extent; 4 = to a great extent; 5 = I don’t know.

**Table 4 foods-09-01811-t004:** Mean scores on the clustering scales and contrast analysis results by clusters of consumers’ motives for eating and choosing SBPs.

	SA:			Swiss:		
	Balanced	Deniers	Frequenters	Balanced	Health-Conscious	Frequenters
	43.4%	21.7%	34.9%	31.8%	30.0%	38.2%
Motives for eating						
Physical eating	2.97	2.42	2.88	2.65	2.35	2.23
Emotional eating ***^,1;^ **^,2^	3.06	2.30	2.71	2.53	1.79	1.88
Environmental eating *^,2^	2.92	2.69	2.76	2.35	2.01	1.73
Motives for choice						
Health ***^,1;^ **^,2^	1.73	2.39	2.80	1.74	1.68	1.33
Mood ***^,1,2^	2.78	2.50	2.51	2.50	1.75	1.58
Convenience ***^,2^	3.54	3.36	3.34	2.76	2.40	1.77
Sensory appeal **^,1,2^	3.79	3.63	3.44	3.51	3.35	2.71
Natural content ***^,1,2^	1.49	2.51	2.70	1.67	2.34	1.51
Price **^,1;^ ***^,2^	2.75	3.24	3.27	2.48	2.72	1.44
Weight control ***^,1^	1.38	1.89	2.61	1.41	1.66	1.13

Note. * *p* < 0.05; ** *p* < 0.01; *** *p* < 0.001. ^1^ SA. ^2^ Swiss. Mean scores on a five-point Likert scale, 1 = never; 2 =almost never; 3 = sometimes; 4 = often; 5 = always.

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
