# Peer review of "Consumers’ Motives for Eating and Choosing Sweet Baked Products: A Cross-Cultural Segmentation Study"

_foods, 2020, doi:10.3390/foods9121811_

Round 1

Reviewer 1 Report

The is an interesting and well written manuscript describing a well-designed research project.

The importance of the study is well established. 

The research may be used for the purpose suggested by the authors, but it can also be used by firms engaged in the production and distribution of UPPs and SBPs to increase sales of such.

Please consider exploring or addressing the following observations/questions.
-Do the relative differences in age, income, and non-work status across the two samples suggest different levels of affluence and exposure to UPP/SBP?
-Is the income distribution within each sample comparable to national averages?
-Can the much higher percentage of paid work in the SA sample compared to the Swiss sample explain some of the findings.  Has relative discretionary income been considered?
-Can the frequency of (prior) consumption explain some of the FCQ choices?  E.g., prior consumption may decrease the identification of SBPs as "treats."
-Is the appearance of "health conscious" in the Swiss sample and not in the SA sample a function of 1st world privilege? 

Reviewer 2 Report

A BRIEF SUMMARY

This manuscript's very interesting containing information about consumers’ motives to eat and choose sweet baked products (SBPs). A cross-cultural segmentation study on a South African (SA) and Swiss population sample 12 (N = 216), was implemented using the Motivation for Eating Scale (MFES) and Food Choice 13 Questionnaire (FCQ). The evaluation methodology for obtaining acceptance data was properly selected and correctly applied. Statistical analysis was also correct. The main contribution of the manuscript is that it provides meaningful information and suggestions to promote healthy food choices and educate consumers about healthy food alternatives. It is evident in this study that the application of presential questionnaires would be necessary because the use of online questionnaires can interfere in obtaining a sample that is effectively representative of the populations. The difference in sample size across the two countries, overall small samples and the different mean ages of the respondents are additional limitations for this study.

The basic weakness of the manuscript is related to the Materials and Methods and Results and Discussion which is explained in detail in the following comments.

The aim of the study:

It is stated in the title, Abstract (Lines 11-14) and Introduction (Lines 61-63).

Introduction:

The Introduction is well written and succinct, supported by a review of recent literature.

Materials and Methods

The legend of Table 1 is not correct (Line 95). The description of Table 1 is not in accordance with the table itself (Lines 83-111). The text refers to four sections but the table itself is not divided into parts. The text of lines 110 and 111 that refers to the fourth section of the table is not actually present in the content of table 1. Therefore, I am of the opinion that point "2.2. Questionnaire" has to be rewritten as well as the table has to be modified to make it more explicit.

Results and Discussion:

I think that's better to combine Results and Discussion because in different sections make reading difficult. This procedure should be applied to the discussion of all tables (Table 2, Table 3 and Table 4).

Round 2

Reviewer 2 Report

The authors reviewed the manuscript considering all of my suggestions.